# RLlib Flow: Distributed Reinforcement Learning is a Dataflow Problem

**Eric Liang**[*]
UC Berkeley

**Zhanghao Wu**[*]
UC Berkeley

**Michael Luo**
UC Berkeley

**Sven Mika**
Anyscale

**Joseph E. Gonzalez**
UC Berkeley

**Ion Stoica**
UC Berkeley

## Abstract

Researchers and practitioners in the field of reinforcement learning (RL) frequently leverage parallel computation, which has led to a plethora of new algorithms and systems in the last few years. In this paper, we re-examine the challenges posed by distributed RL and try to view it through the lens of an old idea: distributed dataflow. We show that viewing RL as a dataflow problem leads to highly composable and performant implementations. We propose RLlib Flow, a hybrid actor-dataflow programming model for distributed RL, and validate its practicality by porting the full suite of algorithms in RLlib, a widely adopted distributed RL library. Concretely, RLlib Flow provides 2-9× code savings in real production code and enables the composition of multi-agent algorithms not possible by end users before. The open-source code is available as part of RLlib at https://github.com/ray-project/ray/tree/master/rllib.

## 1 Introduction

The past few years have seen the rise of deep reinforcement learning (RL) as a new, powerful optimization method for solving sequential decision making problems. As with deep supervised learning, researchers and practitioners frequently leverage parallel computation, which has led to the development of numerous distributed RL algorithms and systems as the field rapidly evolves.

However, despite the high-level of abstraction that RL algorithms are defined in (i.e., as a couple dozen lines of update equations), their implementations have remained quite low level (i.e., at the level of message passing). This is particularly true for *distributed* RL algorithms, which are typically implemented directly on low-level message passing systems or actor frameworks [14]. Libraries such as Acme [15], RLgraph [26], RLlib [19], and Coach [2] provide unified abstractions for defining single-agent RL algorithms, but their user-facing APIs only allow algorithms to execute within the bounds to predefined distributed execution patterns or "templates".

While the aforementioned libraries have been highly successful at replicating a large number of novel RL algorithms introduced over the years, showing the generality of their underlying actor or graph-based computation models, the needs of many researchers and practitioners are often not met by their abstractions. We have observed this firsthand from users of open source RL libraries:

First, RL practitioners are typically not systems engineers. They are not well versed with code that mixes together the logical dataflow of the program and system concerns such as performance and bounding memory usage. This leads to a high barrier of entry for most RL users to experimenting with debugging existing distributed RL algorithms or authoring new distributed RL approaches.

---

∗ indicates equal contributions.

35th Conference on Neural Information Processing Systems (NeurIPS 2021).

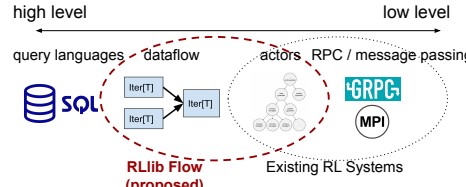

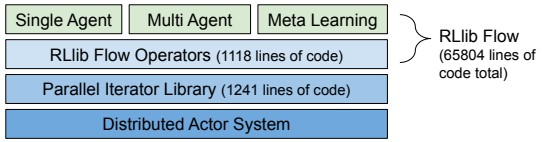

Figure 1: RLlib Flow enables implementation of distributed RL with high-level dataflow.

Figure 2: Architecture of our port of RLlib to RLlib Flow (RLlib Flow core is only 1118 lines of code).

Table 1: Comparison of the systems aspects of RLlib Flow to other distributed RL libraries.

| Distributed Library | Distribution Scheme | Generality | Programmability | #Algorithms |
|---|---|---|---|---|
| RLGraph [26] | Pluggable | General Purpose | Low-level / Pluggable | 10+ |
| Deepmind Acme [15] | Actors + Reverb | Async Actor-Learner | Limited | 10+ |
| Intel Coach [2] | Actors + NFS | Async Actor-Learner | Limited | 30+ |
| RLlib [19] | Ray Actors | General Purpose | Flexible, but Low-level | 20+ |
| **RLlib Flow** | **Actor / Dataflow** | **General Purpose** | **Flexible and High-level** | **20+** |

Second, even when an RL practitioner is happy with a particular algorithm, they may wish to *customize* it in various ways. This is especially important given the diversity of RL tasks (e.g., single-agent, multi-agent, meta-learning). While many customizations within common RL environments can be anticipated and made available as configuration options (e.g., degree of parallelism, batch size), it is difficult for a library author to provide enough options to cover less common tasks that necessarily alter the distributed pattern of the algorithm (e.g., interleaved training of different distributed algorithms, different replay strategies).

Our experience is that when considering the needs of users for novel RL applications and approaches, RL development requires a significant degree of programming flexibility. Advanced users want to tweak or add various distributed components (i.e., they need to write programs). In contrast to supervised learning, it is more difficult to provide a fixed set of abstractions for scaling RL training.

As a result, it is very common for RL researchers or practitioners to eschew existing infrastructure, either sticking to non-parallel approaches, which are inherently easier to understand and customize [3, 6], or writing their own distributed framework that fits their needs. The large number of RL frameworks in existence today is evidence of this, especially considering the number of these frameworks aiming to be "simpler" versions of other frameworks.

In this paper, we re-examine the challenges posed by distributed RL in the light of these user requirements, drawing inspiration from prior work in the field of data processing and distributed dataflow. To meet these challenges, we propose RLlib Flow, a hybrid actor-dataflow model for distributed RL. Like streaming data systems, RLlib Flow provides a small set of operator-like primitives that can be composed to express distributed RL algorithms. Unlike data processing systems, RLlib Flow explicitly exposes references to actor processes participating in the dataflow, permitting limited message passing between them in order to more simply meet the requirements of RL algorithms. The interaction of dataflow and actor messages is managed via special sequencing and concurrency operators.

The contributions of our paper are as follows:

1. We examine the needs of distributed RL algorithms and RL practitioners from a dataflow perspective, identifying key challenges (Section 2 and 3).

2. We propose RLlib Flow, a hybrid actor-dataflow programming model that can simply and efficiently express distributed RL algorithms, and enables composition of multi-agent algorithms not possible by end users before without writing low-level systems code. (Section 4 and 5).

3. We port all the algorithms of a production RL library (RLlib) to RLlib Flow, providing 2-9× savings in distributed execution code, compare its performance with the original implementation, and show performance benefits over systems such as Spark Streaming (Section 6).

## 2 Distributed Reinforcement Learning

We first discuss the relevant computational characteristics of distributed RL algorithms, starting with the common *single-agent training* scenario, where the goal is to optimize a single agent's performance in an environment, and then discuss the computational needs of emerging *multi-agent*, *model-based*, and *meta-learning* training patterns.

### 2.1 RL Algorithm Basics

The goal of an RL algorithm is typically to improve the performance of a *policy* with respect to an objective defined through an *environment* (e.g., simulator). The policy is usually defined as a deep neural network, which can range from several KB to several hundred MB in size. RL algorithms can be generally broken down into the following basic steps:

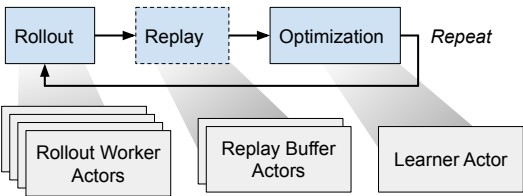

Figure 3: Most RL algorithms can be defined in terms of the basic steps of Rollout, Replay, and Optimization. These steps are commonly parallelized across multiple *actor* processes.

**Rollout**: To generate experiences, the policy, which outputs actions to take given environment observations, is run against the environment to collect batches of data. The batch consists of observations, actions, rewards, and episode terminals and can vary in size (10s to 10000s of steps).

**Replay**: On-policy algorithms (e.g., PPO [27], A3C [22]) collect new experiences from the current policy to learn. On the other hand, off-policy algorithms (e.g., DQN [23], SAC [12]) can leverage experiences from past versions of the policy as well. For these algorithms, a *replay buffer* of past experiences can be used. The size of these buffers ranges from a few hundred to millions of steps.

**Optimization**: Experiences, either freshly collected or replayed, can be used to improve the policy. Typically this is done by computing and applying a gradient update to the policy and value neural networks. While in many applications a single GPU suffices to compute gradient updates, it is sometimes desirable to leverage multiple GPUs within a single node, asynchronous computation of gradients on multiple CPUs [22], or many GPUs spread across a cluster [29].

### 2.2 RL Algorithm Variants

**Single-Agent Training.** Training a single RL agent—the most basic and common scenario—consists of applying the steps of rollout, replay, and optimization repeatedly until the policy reaches the desired performance. Synchronous algorithms such as A2C [22] and PPO apply the steps strictly sequentially. Parallelism may be leveraged internally within each step. Asynchronous algorithm variations such as A3C [22], Ape-X [16], APPO [21], and IMPALA [8], pipeline and overlap the rollout and optimization steps asynchronously to hit higher data throughputs. Rate limiting [15] can be applied to control learning dynamics in the asynchronous setting.

**Multi-Agent Training.** In multi-agent training, there are multiple acting entities in the environment (e.g., cooperating or competing agents). While there is a rich literature on multi-agent algorithms, we note that the *dataflow structure* of multi-agent training is similar to that of single-agent—as long as all entities are being trained with the same algorithm and compatible hyperparameters. However, problems arise should it be required to customize the training of any of the agents in the environment. For example, in a two-agent environment, one agent may desire to be optimized at a higher frequency (i.e., smaller batch size). This fundamentally alters the training dataflow—there are now two iterative loops executing at different frequencies. Furthermore, if these agents are trained with entirely different algorithms, there is a need to compose two different distributed dataflows.

**Model-Based and Meta-Learning Algorithms.** Model-based algorithms seek to learn transition dynamics of the environment to improve the sample efficiency of training. This can be thought

of as adding a supervised training step on top of standard distributed RL, where an ensemble of one or more dynamics models are trained from environment-generated data. Handling the data routing, replay, optimization, and stats collection for these models naturally adds complexity to the distributed dataflow graph, "breaking the mold" of standard model-free RL algorithms and hard to be implemented in low-level systems. Using RLlib Flow, we have implemented two state-of-the-art model-based algorithms: MB-MPO [5] and Dreamer [13].

### 2.3 A Case for a Higher Level Programming Model

Given that existing distributed RL algorithms are already implementable using low level actor and RPC primitives, it is worth questioning the value of defining a higher level computation model. Our experience is that RL is more like data analytics than supervised learning. Advanced users want to tweak or add various distributed components (i.e., they need to program), and there is no way to have a "one size fits all" (i.e., Estimator interface from supervised learning). We believe that, beyond the ability to more concisely and cleanly capture *single-agent* RL algorithms, the computational needs of more advanced RL training patterns motivate higher level programming models like RLlib Flow.

## 3 Reinforcement Learning vs Data Streaming

The key observation behind RLlib Flow is that the dataflow graph of RL algorithms are quite similar to those of data streaming applications. Indeed, RL algorithms can be captured in general purpose dataflow programming models. However, due to several characteristics, they are not a perfect fit, even for dataflow programming models that support iterative computation.

In this section we examine the dataflow of the A3C algorithm (Figure 4) to compare and contrast RL with streaming dataflow. A3C starts with (1) parallel rollouts across many experiences. Policy gradients are computed in parallel based on rollouts in step (2). In step (3), the gradients are asynchronously gathered and applied on a central model, which is then used to update rollout worker weights. Importantly, each box or *operator* in this dataflow may be *stateful* (e.g., `ParallelRollouts` holds environment state as well as the current policy snapshot).

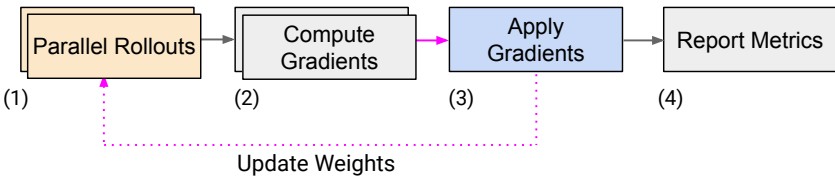

Figure 4: The dataflow of the A3C parallel algorithm. Each box is an operator or iterator from which data items can be pulled from. Here operators (1) and (2) represent parallel computations, but (3) and (4) are sequential. Black arrows denote synchronous data dependencies, pink arrows asynchronous dependencies, and dotted arrows actor method calls.

Similar to data processing topologies, A3C is applying a transformation to a data stream (of rollouts) in parallel (to compute gradients). This is denoted by the black arrow between (1) and (2). There is also a non-parallel transformation to produce metrics from the computation, denoted by the black arrow between (3) and (4) However, zooming out to look at the entire dataflow graph, a few differences emerge:

**Asynchronous Dependencies**: RL algorithms often leverage asynchronous computation to reduce update latencies and eliminate stragglers [22]. In RLlib Flow, we represent these with a pink arrow between a parallel and sequential iterator. This means items will be fetched into the sequential iterator as soon as they are available, instead of in a deterministic ordering. The level of asynchrony can be configured to increase pipeline parallelism.

**Message Passing**: RL algorithms, like all iterative algorithms, need to update upstream operator state during execution (e.g., update policy weights). Unlike iterative algorithms, these updates may be fine-grained and asynchronous (i.e., update the parameters of a particular worker), as well as coarse-grained (i.e., update all workers at once after a global barrier). RLlib Flow allows method

calls (messages) to be sent to any actor in the dataflow. Ordering of messages in RLlib Flow with respect to dataflow steps is guaranteed if synchronous data dependencies (black arrows) fully connect the sender to the receiver, providing *barrier semantics*.

**Consistency and Durability**: Unlike data streaming, which has strict requirements such as exactly-once processing of data [30], RL has less strict consistency and durability requirements. This is since on a fault, the entire computation can be restarted from the last checkpoint with minimal loss of work. Message or data loss can generally be tolerated without adverse affect on training. Individual operators can be restarted on failure, discarding any temporary state. This motivates a programming model that minimizes overhead (e.g., avoids state serialization and logging cost).

## 4    A Dataflow Programming Model for Distributed RL

Here we formally define the RLlib Flow hybrid actor-dataflow programming model. RLlib Flow consists of a set of dataflow operators that produce and consume *distributed iterators* [11]. These distributed iterators can represent parallel streams of a data items T sharded across many actors (`ParIter[T]`), or a single sequential stream of items (`Iter[T]`). It is important to note that these iterators are *lazy*, they do not execute computation or produce items unless requested. This means that the entire RLlib Flow execution graph driven by taking items from the output operator.

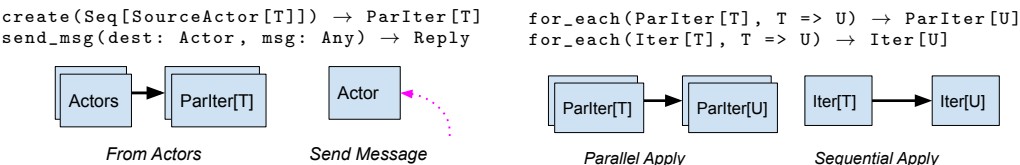

```
create(Seq[SourceActor[T]]) → ParIter[T]
send_msg(dest: Actor, msg: Any) → Reply
```

*From Actors*          *Send Message*

Figure 5: Creation and Message Passing

```
for_each(ParIter[T], T => U) → ParIter[U]
for_each(Iter[T], T => U) → Iter[U]
```

*Parallel Apply*          *Sequential Apply*

Figure 6: Transformation

**Creation and Message Passing**: RLlib Flow iterators are always created from an existing set of actor processes. In Figure 4, the iterator is created from a set of rollout workers that produce experience batches given their current policy. Also, any operator may send a message to any source actor (i.e., a rollout worker, or replay buffer) during its execution. In the A3C example, the update weights operation is a use of this facility. The order guarantees of these messages with respect to dataflow steps depends on the barrier semantics provided by *sequencing operators*. The sender may optionally block and await the reply of sent messages. We show the operator in Figure 5.

**Transformation**: As in any data processing system, the basic operation of data transformation is supported. Both parallel and sequential iterators can be transformed with the `for_each` operator. The transformation function can be stateful (i.e., in Python it can be a callable function class that holds state in class members, and in the case of sequential operators it can reference local variables via closure capture). In the A3C example, `for_each` is used to compute gradients for each batch of experiences, which depends on the current policy state of the source actor. In the case of the `ComputeGradients` step, this state is available in the local process memory of the rollout worker, and is accessible because RLlib Flow schedules the execution of parallel operations onto the source actors. We show the operator in Figure 6.

```
gather_async(ParIter[T],
             num_async: Int) → Iter[T]
gather_sync(ParIter[T]) → Iter[List[T]]
next(Iter[T]) → T
```

*Async Gather*          *Bulk Sync Gather*
*(No Barrier)*          *(Full Barrier)*

Figure 7: Sequencing

```
split(Iter[T]) → (Iter[T], Iter[T])
union(List[Iter[T]],
      weights: List[float]) → Iter[T]
union_async(List[Iter[T]]): Iter[T]
```

*Duplicate*          *Union*          *Async Union*

Figure 8: Concurrency

**Sequencing**: To consume a parallel iterator, the items have to be serialized into some sequential order. This is the role of sequencing operators. Once converted into a sequential iterator, `next` can

be called on the iterator to fetch a concrete item from the iterator. The `gather_async` operator is used in A3C, and gathers computed gradients as fast as they are computed for application to a central policy. For a deterministic variation, we could have instead used `gather_sync`, which waits for one gradient from each shard of the iterator before returning. The sync gather operator also has *barrier semantics*. Upstream operators connected by a synchronous dependencies (black arrows) are fully halted between item fetches. This allows for the source actors to be updated prior to the next item fetch. Barrier semantics do not apply across asynchronous dependencies, allowing the mixing of synchronous and async dataflow fragments separated by pink arrows, in Figure 7.

**Concurrency**: Complex algorithms may involve multiple concurrently executing dataflow fragments. Concurrency (`union`) operators, in Figure 8, govern how these concurrent iterators relate to each other. For example, one may wish two iterators to execute sequentially in a round robin manner, execute independently in parallel, or rate limiting progress to a fixed ratio [15]. Additionally, one might wish to duplicate (`split`) an iterator, in which case buffers are automatically inserted to retain items until fully consumed. In this case, the RLlib Flow scheduler tries to bound memory usage by prioritizing the consumer that is falling behind.

## 5 Implementation

We implemented RLlib Flow on the Ray distributed actor framework [24] as two separate modules: a general purpose parallel iterator library (1241 lines of code), and a collection of RL specific dataflow operators (1118 lines of code) (Figure 2). We then ported the full suite of 20+ RL algorithms in RLlib to RLlib Flow, replacing the original implementations built directly on top of low-level actor and RPC primitives. Only the portions of code in RLlib related to distributed execution were changed (the exact same numerical computations are run in our port), which allows us to fairly evaluate against it as a baseline. In this section we overview two simple examples to illustrate RLlib Flow. MAML case study, can be found in Section A.2.

### 5.1 Asynchronous Optimization in RLlib Flow vs RLlib

As previously seen in Figure 4, A3C is straightforward to express in RLlib Flow. Figure 9a shows pseudocode for A3C in RLlib Flow (11 lines), which we compare to a simplified version of the RLlib implementation (originally 87 lines). RLlib Flow hides the low-level worker management and data communication with its dataflow operators, providing more readable and flexible code. More detailed comparison of implementations in RLlib Flow and RLlib can be found in Section A.3.

```
1 # type: List[RolloutActor]
2 workers = create_rollout_workers()
3 # type: Iter[Gradients]
4 grads = ParallelRollouts(workers)
5     .par_for_each(ComputeGradients())
6     .gather_async()
7 # type: Iter[TrainStats]
8 apply_op = grads
9     .for_each(ApplyGradients(workers))
10 # type: Iter[Metrics]
11 return ReportMetrics(apply_op, workers)
```

```
1 # launch gradients computation tasks
2 pending_gradients = dict()
3 for worker in remote_workers:
4     worker.set_weights.remote(weights)
5     future = worker.compute_gradients
6         .remote(worker.sample.remote())
7     pending_gradients[future] = worker
8 # asynchronously gather gradients and apply
9 while pending_gradients:
10    wait_results = ray.wait(
11        pending_gradients.keys(),
12        num_returns=1)
13    ready_list = wait_results[0]
14    future = ready_list[0]
15
16    gradient, info = ray.get(future)
17    worker = pending_gradients.pop(future)
18    # apply gradients
19    local_worker.apply_gradients(gradient)
20    weights = local_worker.get_weights()
21    worker.set_weights.remote(weights)
22    # launch gradient computation again
23    future = worker.compute_gradients
24        .remote(worker.sample.remote())
25    pending_gradients[future] = worker
```

(a) The entire A3C dataflow in RLlib Flow.      (b) A small portion of the RLlib A3C policy optimizer.

Figure 9: Comparing the implementation of asynchronous optimization in RLlib Flow vs RLlib.

## 5.2 Ape-X Prioritized Experience Replay in RLlib Flow

Ape-X [17] (Figure 10a) is a high-throughput variation of DQN. It is notable since it involves multiple concurrent sub-flows (experience storage, experience replay), sets of actors (rollout actors, replay actors), and actor messages (updating model weights, updating replay buffer priorities). The sub-flows (`store_op`, `replay_op`) can be composed in RLlib Flow as follows using the `Union` operator (Figure 10b). The complicated workflow can be implemented in several lines, as shown in Figure 10b.

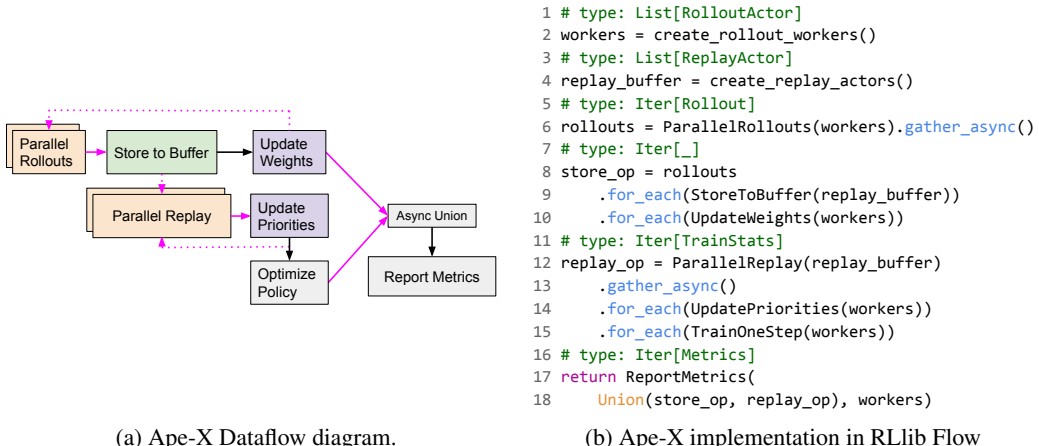

```
1  # type: List[RolloutActor]
2  workers = create_rollout_workers()
3  # type: List[ReplayActor]
4  replay_buffer = create_replay_actors()
5  # type: Iter[Rollout]
6  rollouts = ParallelRollouts(workers).gather_async()
7  # type: Iter[_]
8  store_op = rollouts
9      .for_each(StoreToBuffer(replay_buffer))
10     .for_each(UpdateWeights(workers))
11 # type: Iter[TrainStats]
12 replay_op = ParallelReplay(replay_buffer)
13     .gather_async()
14     .for_each(UpdatePriorities(workers))
15     .for_each(TrainOneStep(workers))
16 # type: Iter[Metrics]
17 return ReportMetrics(
18     Union(store_op, replay_op), workers)
```

(a) Ape-X Dataflow diagram.  (b) Ape-X implementation in RLlib Flow

Figure 10: Dataflow and implementation for Ape-X algorithm. Two dataflow fragments are executed concurrently to optimize the policy.

## 5.3 Composing DQN and PPO in Multi-Agent Training

Multi-agent training can involve the composition of different training algorithms (i.e., PPO and DQN). Figure 11a shows the combined dataflow for an experiment that uses DQN to train certain policies in an environment and PPO to train others. The code can be found in Figure 11b. In an actor or RPC-based programming model, this type of composition is difficult because dataflow and control flow logic is intermixed. However, it is easy to express in RLlib Flow using the `Union` operator (Figure 8). In Figure 12, we show the implementation of the two subflow, `ppo_plan` and `dqn_plan`, in the multi-agent training (Figure 11b).

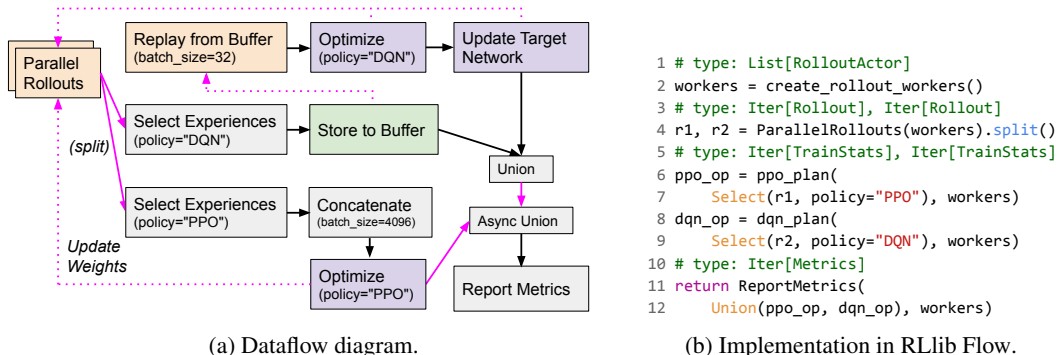

```
1  # type: List[RolloutActor]
2  workers = create_rollout_workers()
3  # type: Iter[Rollout], Iter[Rollout]
4  r1, r2 = ParallelRollouts(workers).split()
5  # type: Iter[TrainStats], Iter[TrainStats]
6  ppo_op = ppo_plan(
7      Select(r1, policy="PPO"), workers)
8  dqn_op = dqn_plan(
9      Select(r2, policy="DQN"), workers)
10 # type: Iter[Metrics]
11 return ReportMetrics(
12     Union(ppo_op, dqn_op), workers)
```

(a) Dataflow diagram.  (b) Implementation in RLlib Flow.

Figure 11: Dataflow and implementation for concurrent multi-agent multi-policy workflow with PPO and DQN agents in an environment.

## 6 Evaluation

In our evaluation, we seek to answer the following questions:

```
1 ppo_op = rollouts
2     .for_each(Select(policy="PPO"))
3     .combine(ConcatBatches(batch_size))
4     .for_each(TrainOneStep(workers))
```

```
1 replay_buffer = create_replay_actors()
2 store_op = rollouts
3     .for_each(Select(policy="DQN"))
4     .for_each(StoreToBuffer(replay_buffer))
5 replay_op = Replay(replay_buffer)
6     .for_each(TrainOneStep(workers))
7     .for_each(UpdateTargetNetwork(workers))
8 dqn_op = Union(store_op, replay_op)
```

(a) Implementation of PPO Subflow.                    (b) Implementation of DQN Subflow.

Figure 12: Implementation of sub-flows for the multi-agent multi-policy training of PPO and DQN.

1. What is the quantitative improvement in code complexity with RLlib Flow?

2. How does RLlib Flow compare to other systems in terms of flexibility and performance for RL tasks?

## 6.1   Code Complexity

**Lines of Code**: In Table 2 we compare the original algorithms in RLlib to after porting to RLlib Flow. No functionality was lost in the RLlib Flow re-implementations. We count all lines of code directly related to distributed execution, including comments and instrumentation code, but not including utility functions shared across all algorithms. For completeness, for RLlib Flow we include both an minimal (`RLlib Flow`) and conservative (`+shared`) estimate of lines of code. The conservative estimate includes lines of code in shared operators. Overall, we observe between a 1.9-9.6× (optimistic) and 1.1-3.1× (conservative) reduction in lines of code with RLlib Flow. The most complex algorithm (IMPALA) shrunk from 694 to 89-362 lines.

Table 2: Lines of code for several prototypical algorithms implemented with the original RLlib vs our RLlib Flow-based RLlib. *Original MAML: https://github.com/jonasrothfuss/ProMP*

|        | RLlib | RLlib Flow | +shared | Ratio     |
|--------|-------|------------|---------|-----------|
| A3C    | 87    | 11         | 52      | 1.6-9.6×  |
| A2C    | 154   | 25         | 50      | 3.1-6.1×  |
| DQN    | 239   | 87         | 139     | 1.7-2.7×  |
| PPO    | 386   | 79         | 225     | 1.7-4.8×  |
| Ape-X  | 250   | 126        | 216     | 1.1-1.9×  |
| IMPALA | 694   | 89         | 362     | 1.9-7.8×  |
| MAML   | 370*  | 136        | 136     | 2.7×      |

**Readability**: We believe RLlib Flow provides several key benefits for readability of RL algorithms:

1. The high-level dataflow of an algorithm is visible at a glance in very few lines of code, allowing readers to understand and modify execution pattern without diving deep into the execution logic.

2. Execution logic is organized into individual operators, each of which has a consistent input and output interface (i.e., transforms an iterator into another iterator). In contrast to building on low-level RPC systems, developers can decompose their algorithms into reusable operators.

3. Performance concerns are isolated into the lower-level parallel iterator library. Developers do not need to deal with low-level concepts such as batching or flow-control.

**Flexibility**: As evidence of RLlib Flow's flexibility, an undergraduate was able to implement several model-based (e.g., MB-MPO) and meta-learning algorithms (e.g., MAML), neither of which fit into previously existing execution patterns in RLlib Flow. This was only possible due to the flexibility of RLlib Flow's model. RLlib Flow captures MAML in 139 lines compared to a baseline of ≈370 lines (Table 2). Detailed discussion can be found in Section A.2.1.

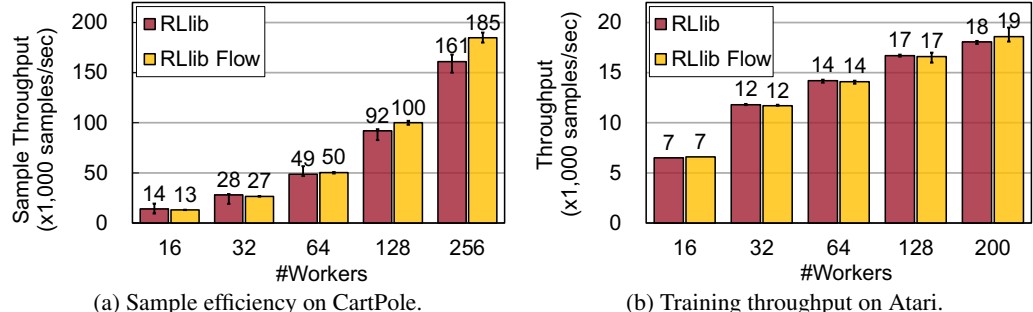

(a) Sample efficiency on CartPole.          (b) Training throughput on Atari.

Figure 13: Performance of RLlib compared with RLlib Flow, executing identical numerical code. RLlib Flow achieves comparable or better performance across different environments.

## 6.2 Microbenchmarks and Performance Comparisons

For all the experiments, we use a cluster with an AWS p3.16xlarge GPU head instance with additional m4.16xlarge worker instances. All machines have 64 vCPUs and are connected by a 25Gbps network. More experiments for different RL algorithms can be found in https://github.com/ray-project/rl-experiments.

**Sampling Microbenchmark**: We evaluate the data throughput of RLlib Flow in isolation by running RL training with a dummy policy (with only one trainable scalar). Figure 13a shows that RLlib Flow achieves slightly better throughput due to small optimizations such as batched RPC wait, which are easy to implement across multiple algorithms in a common way in RLlib Flow.

**IMPALA Throughput**: In Figure 13b we benchmark IMPALA, one of RLlib's high-throughput RL algorithms, and show that RLlib Flow achieves similar or better end-to-end performance.

**Performance of Multi-Agent Multi-Policy Workflow**: In Figure 14, we show that the workflow of the two-trainer example (Figure 11a) achieves close to the theoretical best performance possible combining the two workflows (calculated via Amdahl's law). This benchmark was run in a multi-agent Atari environment with four agents per policy, and shows RLlib Flow can be practically used to compose complex training workflows.

**Comparison to Spark Streaming**: Distributed dataflow systems such as Spark Streaming [30] and Flink [1] are designed for collecting and transforming live data streams from online applications (e.g., event streams, social media). Given the basic *map* and *reduce* operations, we can implement synchronous RL algorithms in any of these streaming frameworks. However, without consideration for the requirements of RL tasks (Section 3), these frameworks can introduce significant overheads. In Figure 15 we compare the performance of PPO implemented in Spark Streaming and RLlib Flow. Implementation details are in in Appendix A.1.

## 7 Related Work

**Reinforcement Learning Systems**: RLlib Flow is implemented concretely in RLlib, however, we hope it can provide inspiration for a new generation of general purpose RL systems. RL libraries available today range from single-threaded implementations [4, 6, 7, 18] to distributed [2, 9, 15, 19, 20, 26]. These libraries often focus on providing common frameworks for the numerical concerns of RL algorithms (e.g., loss, exploration, and optimization steps).

However, these aforementioned libraries rely on *predefined* distributed execution patterns. For example, for the Ape-X dataflow in Figure 10a, RLlib defines this with a fixed "AsyncReplayOptimizer"* class that implements the topology, intermixing the dataflow and the control flow.; RLGraph uses an adapted implementation† from RLlib as part of their Ape-X algorithm meta-graph, while Coach does not support Ape-X‡. These execution patterns are predefined as they are low-level, complex to implement, and cannot be modified using high-level end-user APIs. In contrast, RLlib Flow proposes

---

*https://docs.ray.io/en/releases-0.7.7/_modules/ray/rllib/optimizers/async_replay_optimizer.html
†https://github.com/rlgraph/rlgraph/blob/master/rlgraph/execution/ray/apex/apex_executor.py
‡https://github.com/IntelLabs/coach

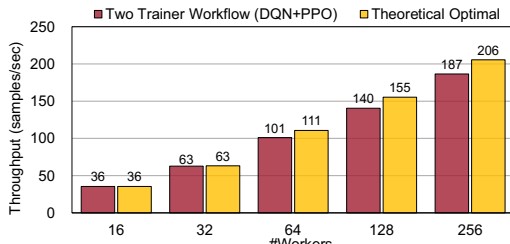

Figure 14: RLlib Flow achieves close to the theoretical optimal performance combining two workflows for multi-agent training, making it practical to use for composing complex training scenarios.

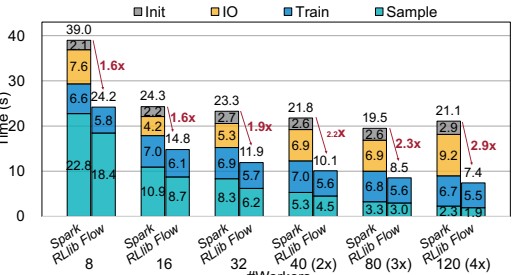

Figure 15: The throughput comparison between RLlib Flow and Spark Streaming with PPO algorithm on CartPole-v0 environment.

a high-level distributed programming model for RL algorithm implementation, exposing this pattern in much fewer lines of code (Figure 10b), and allowing free composition of these patterns by users (Figure 11b). The ideas from RLlib Flow can be integrated with any RL library to enable flexibility in distributed execution.

**Distributed Computation Models**: RLlib Flow draws inspiration from both streaming dataflow and actor-based programming models. Popular open source implementations of streaming dataflow, including Apache Storm [28], Apache Flink [1], and Apache Spark [30, 31] transparently distribute data to multiple processors in the background, hiding the scheduling and message passing for distribution from programmers. In Appendix A.1, we show how distributed PPO can be implemented in Apache Spark. Apache Flink's `Delta Iterate` operator can similarly support synchronous RL algorithms. However, data processing frameworks have limited asynchronous iteration support.

The Volcano model [11], commonly used for distributed data processing, pioneered the parallel iterator abstraction. RLlib Flow builds on the Volcano model to not only encapsulate parallelism, but also to encapsulate the synchronization requirements between concurrent dataflow fragments, enabling users to also leverage actor message passing.

Naiad [25] is a low-level distributed dataflow system that supports cyclic execution graphs and message passing. It is designed as a system for implementing higher-level programming models. In principle, it is possible to implement the RLlib Flow model in Naiad. Transformation operators can be placed on the stateful vertices of the execution graph. The message passing and concurrency (`Union`) operators can be represented by calling SENDBY and ONRECV interface on senders and receivers, which support asynchronous execution. RLlib Flow's barrier semantics can be expressed with ONNOTIFY and NOTIFYAT, where the former indicates all the required messages are ready, and the latter blocks execution until the notification has been received. We implemented RLlib Flow on Ray instead of Naiad for practical reasons (e.g., Python support).

## 8 Conclusion

In summary, we propose RLlib Flow, a hybrid actor-dataflow programming model for distributed RL. We designed RLlib Flow to simplify the understanding, debugging, and customization of distributed RL algorithms RL developers require. RLlib Flow provides comparable performance to reference algorithms implemented directly on low-level actor and RPC primitives, enables complex multi-agent and meta-learning use cases, and reduces the lines of code for distributed execution in a production RL library by 2-9×. RLlib Flow is available as part of the open source RLlib project, and we hope it can also help inform the design of future RL libraries.

## 9 Acknowledgement

In addition to NSF CISE Expeditions Award CCF-1730628, this research is supported by gifts from Amazon Web Services, Ant Group, Ericsson, Facebook, Futurewei, Google, Intel, Microsoft, Scotiabank, and VMware.

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
