## A.1 RL in Spark Streaming

**PPO Implementation.** In Figure A1, we show the high-level pseudocode of our port of the PPO algorithm to Spark Streaming. Similar to our port of RLlib to RLlib Flow, we only changed the parts of the PPO algorithm in RLlib that affect distributed execution, keeping the core algorithm implementation (e.g., numerical definition of policy loss and neural networks in TensorFlow) as similar as possible for fair comparison. We made a best attempt at working around aforementioned limitations (e.g., using a `binaryRecordsStream` input source to efficiently handle looping, defining efficient serializers for neural network state, and adjusting the microbatching to emulate the RLlib configuration).

```
1  # RL on Spark Streaming:
2  # Iterate by saving/detecting states file in a folder:
3  #   1) Replicate the states to workers
4  #   2) Sample in parallel (map)
5  #   3) Collect the samples (reduce)
6  #   4) Train on sampled batch
7  #   5) Save the states and trigger next iteration
8
9  # Set up the Spark cluster
10 sc = SparkContext(master_addr)
11 # Spark detects new states file in path
12 states = sc.binaryRecordsStream(path)
13 rep = states.flatMap(replicate_fn)
14 split = rep.repartion(NUM_WORKERS)
15 # Restore actor from states and sample
16 sample = splits.map(actor_sample_fn)
17 # Collect all samples from actors
18 reduced = sample.reduce(merge_fn)
19 # Restore trainer from states and train
20 new_states = reduced.map(train_fn)
21 # Save sampling/training states to path
22 new_states.foreachRDD(save_states_fn)
```

Figure A1: Example of Spark Streaming for Distributed RL.

**Experiment Setup.** We conduct comparisons between the performance of both implementations. In the experiment, we adopt the PPO algorithm for the CartPole-v0 environment with a fixed sampling batch size $B$ of 100K. Each worker samples ($B/\#$ workers) samples each iteration, and for simplicity, the learner updates the model on CPU using a minibatch with 128 samples from the sampled batch. Experiments here are conducted on AWS m4.10xlarge instances.

**Data Framework Limitations**: Spark Streaming is a data streaming framework designed for general purpose data processing. We note several challenges we encountered attempting to port RL algorithms to Spark Streaming:

1. Support for asynchronous operations. Data processing systems like Spark Streaming do not support asynchronous or non-deterministic operations that are needed for asynchronous RL algorithms.
2. Looping operations are not well supported. While many dataflow models in principle support iterative algorithms, we found it necessary to work around them due to lack of language APIs (i.e., no Python API).
3. Support for non-serializable state. In the dataflow model, there is no way to persist arbitrary state (i.e., environments, neural network models on the GPU). While necessary for fault-tolerance, the requirement for serializability impacts the performance and feasibility of many RL workloads.
4. Lack of control over batching. We found that certain constructs such as the data batch size for on-policy algorithms are difficult to control in traditional streaming frameworks, since they are not part of the relational data processing model.

For a single machine (the left three pairs), the breakdown of the running time indicates that the initialization and I/O overheads slow down the training process for Spark comparing to our RLlib Flow. The former overheads come from the nature of Spark that the transformation functions do not persist variables. We have to serialize both the sampling and training states and re-initialize the variables in the next iteration to have a continuous running process. On the other hand, the I/O overheads come from looping back the states back to the input. As an event-time driven streaming system, the stream engine detects changes for the saved states from the source directory and starts new stream processing. The disk I/O leads to high overheads compared to RLlib Flow.

For distributed situation (the right three pairs), the improvement of RLlib Flow becomes more significant against Spark, up to 2.9×. As the number of workers scales up, the sampling time decreases for both the dataflow model. Still, the initialization and I/O overheads stay unchanged, leading to lesser scalability for Spark.

## A.2 Implementation Examples

### A.2.1 Example: MAML

Figure A2b concisely expresses MAML's dataflow (also shown in Figure A2a) [10]. The MAML dataflow involves nested optimization loops; workers collect pre-adaptation data, perform inner adaptation (i.e., individual optimization calls to an ensemble of models spread across the workers), and collect post-adaptation data. Once inner adaptation is complete, the accumulated data is batched together to compute the meta-update step, which is broadcast to all workers.

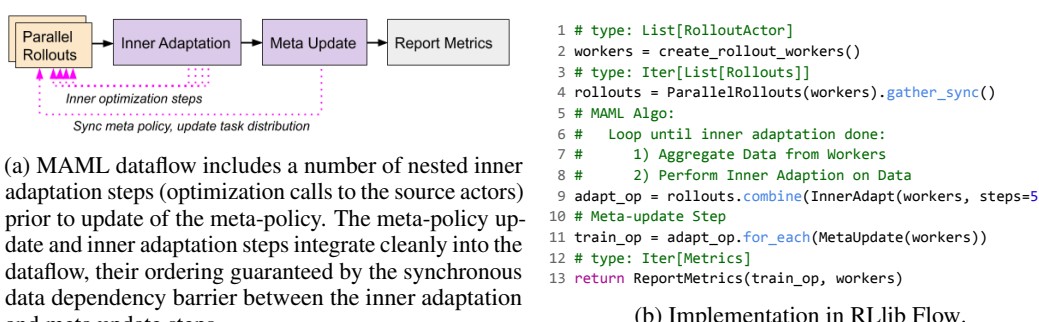

(a) MAML dataflow includes a number of nested inner adaptation steps (optimization calls to the source actors) prior to update of the meta-policy. The meta-policy update and inner adaptation steps integrate cleanly into the dataflow, their ordering guaranteed by the synchronous data dependency barrier between the inner adaptation and meta update steps.

```
1  # type: List[RolloutActor]
2  workers = create_rollout_workers()
3  # type: Iter[List[Rollouts]]
4  rollouts = ParallelRollouts(workers).gather_sync()
5  # MAML Algo:
6  #   Loop until inner adaptation done:
7  #       1) Aggregate Data from Workers
8  #       2) Perform Inner Adaption on Data
9  adapt_op = rollouts.combine(InnerAdapt(workers, steps=5))
10 # Meta-update Step
11 train_op = adapt_op.for_each(MetaUpdate(workers))
12 # type: Iter[Metrics]
13 return ReportMetrics(train_op, workers)
```

(b) Implementation in RLlib Flow.

Figure A2: Dataflow and implementation of the MAML algorithm.

## A.3 Comparison of Implementations in RLlib Flow and RLlib

In this section we report the detailed code comparison of our RLlib Flow and the original RLlib. Listing A1 and Listing A2 are the detailed implementation of A3C in RLlib Flow and RLlib, respectively. Note that the detailed implementation in Listing A1 is exactly the same as we shown before in Figure 9a, but RLlib implementation is much more complicated as the intermixing of the control and data flow. In Listing A3 and Listing A4, we also show the detailed implementation of Ape-X algorithm in our RLlib Flow and RLlib respectively, which also indicates the simplicity, readability and flexibility of our RLlib Flow.

Listing A1: Detailed A3C in RLlib Flow.

```
1  # type: List[RolloutActor]
2  workers = create_rollout_workers()
3  # type: Iter[Gradients]
4  grads = ParallelRollouts(workers)
5      .par_for_each(ComputeGradients())
6      .gather_async()
7  # type: Iter[TrainStats]
8  apply_op = grads
9      .for_each(ApplyGradients(workers))
10 # type: Iter[Metrics]
11 return ReportMetrics(apply_op, workers)
```

Listing A2: Detailed A3C in original RLlib.

```python
1  # Create timers
2  apply_timer = TimerStat()
3  wait_timer = TimerStat()
4  dispatch_timer = TimerStat()
5
6  # Create training information
7  num_steps_sampled = 0
8
9  # type: List[RolloutActor]
10 workers = create_rollout_workers()
11
12 # Get weights from the local rollout actor
13 local_worker = workers.local_worker()
14 weights = local_worker.get_weights()
15
16 # Put weights in raylet (distributed storage)
17 weights = ray.put(weights)
18
19 # type: Dict[obj_id, RolloutActor]
20 pending_gradients = dict()
21
22 # Get the remote rollout actors
23 remote_worker = workers.remote_workers()
24
25 # Issue gradient computation tasks
26 for worker in remote_worker:
27     # Set weight on remote rollout actor
28     worker.set_weights.remote(weights)
29     # Collect samples from the remote rollout actor
30     samples = worker.sample.remote()
31
32     # Kick off gradient computation
33     future = worker.compute_gradients.remote(samples)
34
35     # Map the object id to rollout actor
36     pending_gradients[future] = worker
37
38 # Start training loop
39 while pending_gradients:
40     # Record the time to wait gradient
41     with wait_timer:
42         # Get the list of the futures
43         futures = list(pending_gradients.keys())
44
45         # Wait for one actor to complete
46         wait_results = ray.wait(futures,
47                                 num_returns=1)
48
49         # Get the ready future
50         ready_list = wait_results[0]
51         future = ready_list[0]
52
53         # Get and free the gradient and training infos
```

```
54          # from the raylet (maybe on the remote worker)
55          gradient, info = ray_get_and_free(future)
56
57          # Pop the used gradient from the map
58          worker = pending_gradients.pop(future)
59
60      # Check the validation of the gradient
61      if gradient is not None:
62          # Record the time for gradient apply
63          with apply_timer:
64              # Apply the gradient on the local worker
65              local_worker = workers.local_worker()
66              local_worker.apply_gradients(gradient)
67
68          # Record the metrics from the worker
69          num_steps_sampled += info["batch_count"]
70
71      # Record the time to set new weight on the worker
72      # and launch gradient computation task
73      with dispatch_timer:
74          # Get the weight on local rollout actor
75          local_worker = workers.local_worker()
76          weights = local_worker.get_weights()
77
78          # Set weight on the rollout actor
79          worker.set_weights.remote(weights)
80
81          # Sample rollouts on the rollout actor
82          samples = worker.sample.remote()
83          # Launch gradient computation task on the worker
84          future = worker.compute_gradients.remote(samples)
85
86          # Map the new object id to the corresponding worker
87          pending_gradients[future] = worker
```

Listing A3: Detailed Ape-X in RLlib Flow.

```
1  # type: List[RolloutActor]
2  workers = create_rollout_workers()
3
4  # Create a number of replay buffer actors.
5  replay_actors = create_colocated(ReplayActor)
6
7  # Start the learner thread.
8  learner_thread = LearnerThread(workers.local_worker())
9  learner_thread.start()
10
11 # We execute the following steps concurrently:
12 # (1) Generate rollouts and store them in our replay buffer actors. Update
13 # the weights of the worker that generated the batch.
14 rollouts = ParallelRollouts(workers, mode="async", num_async=2)
15 store_op = rollouts \
```

```
16        .for_each(StoreToReplayBuffer(actors=replay_actors))

17

18  # Only need to update workers if there are remote workers.
19  store_op = store_op.zip_with_source_actor() \
20        .for_each(UpdateWorkerWeights(workers))

21

22  # (2) Read experiences from the replay buffer actors and send to the
23  # learner thread via its in-queue.
24  replay_op = Replay(actors=replay_actors, num_async=4) \
25        .zip_with_source_actor() \
26        .for_each(Enqueue(learner_thread.inqueue))

27

28  # (3) Get priorities back from learner thread and apply them to the
29  # replay buffer actors.
30  update_op = Dequeue(learner_thread.outqueue) \
31        .for_each(UpdateReplayPriorities()) \
32        .for_each(TrainOneStep(workers))

33

34  # Execute (1), (2), (3) asynchronously as fast as possible. Only output
35  # items from (3) since metrics aren't available before then.
36  merged_op = Concurrently(
37        [store_op, replay_op, update_op], mode="async", output_indexes=[2])

38

39  return ReportMetrics(merged_op, workers)
```

Listing A4: Detailed Ape-X in original RLlib. We leave out some of the configurable argument for simplicity.

```
1  # type: List[RolloutActor]
2  workers = create_rollout_workers()

3

4  # Create a learner thread in the main driver to handle
5  # the asynchronous training
6  local_worker = workers.local_worker()
7  learner = LearnerThread(local_worker)

8

9  # Start the learner thread and wait for the input
10  learner.start()

11

12  # Create replay actor handling the replay buffer
13  # create_located: create multiple colocated replay actor
14  # in the same machine as main driver
15  replay_actors = create_colocated(ReplayActor)

16

17  # Create timers
18  timers = {
19        k: TimerStat()
20        for k in [
21              "put_weights", "get_samples", "sample_processing",
22              "replay_processing", "update_priorities", "train", "sample"
23        ]
24  }
```

```python
25
26 # Create training information
27 num_weight_syncs = 0
28 num_samples_dropped = 0
29 learning_started = False
30
31 # Number of worker steps since the last weight update
32 steps_since_update = dict()
33
34 # Create manager for replay
35 replay_tasks = TaskPool()
36 # Kick off replay tasks for local gradient updates
37 for actor in replay_actors:
38     # Start replay task on remote replay actors
39     for _ in range(REPLAY_QUEUE_DEPTH):
40         replay_task = actor.replay.remote()
41         # add replay task into the manager
42         replay_tasks.add(actor, replay_task)
43
44 # Create manager for sampling
45 sample_tasks = TaskPool()
46
47 # Get weights of local worker
48 local_worker = workers.local_worker()
49 weights = local_worker.get_weights()
50
51 # Kick off async background sampling and set the weights
52 # on remote rollout actors
53 remote_workers = workers.remote_workers()
54 for worker in remote_workers:
55     # Set weights
56     worker.set_weights.remote(weights)
57     # Initialize training info for the rollout actor
58     steps_since_update[worker] = 0
59     for _ in range(SAMPLE_QUEUE_DEPTH):
60         # Start sample_with_count task on remote worker
61         sample_with_count_task = worker.sample_with_count.remote()
62         # Add task in to the sample task manager
63         sample_tasks.add(worker, sample_with_count_task)
64
65 # Optimize the model for one step
66 def step(self):
67     # Check the availability of the asynchronous learner thread
68     # and the remote rollout actors
69     assert self.learner.is_alive()
70     assert len(self.workers.remote_workers()) > 0
71
72     # Record the start time for training info
73     start = time.time()
74
75     # Create variables for training
76     sample_timesteps, train_timesteps = 0, 0
77     weights = None
78
```

```python
79          # Record the sampling and processing step
80          with timers["sample_processing"]:
81              # Check the completed sampling task in the sampling manager (TaskPool)
82              completed = list(sample_tasks.completed())
83
84              # Gather the train info, counts of samples
85              counts = ray_get_and_free([c[1][1] for c in completed])
86
87              # Update training information and weights
88              for i, (worker, (sample_batch, count)) in enumerate(completed):
89                  # Update training information
90                  sample_timesteps += counts[i]
91
92                  # Randomly choose one replay actor and send data to it
93                  random_replay_actor = random.choice(replay_actors)
94                  random_replay_actor.add_batch.remote(sample_batch)
95
96                  # Update train info
97                  steps_since_update[worker] += counts[i]
98
99                  # Update weights on remote rollout worker if needed
100                 if steps_since_update[worker] >= MAX_WEIGHT_SYNC_DELAY:
101                     # Note that it's important to pull new weights once
102                     # updated to avoid excessive correlation between actors
103                     if weights is None or learner.weights_updated:
104                         learner.weights_updated = False
105
106                         # Record time for putting weights
107                         with timers["put_weights"]:
108                             # Put local weights in raylet
109                             local_worker = workers.local_worker()
110                             local_weights = local_worker.get_weights()
111                             weights = ray.put(local_weights)
112
113                     # Set weights on the remote rollout worker
114                     worker.set_weights.remote(weights)
115
116                     # Update train info
117                     num_weight_syncs += 1
118                     steps_since_update[worker] = 0
119
120                 # Kick off another sample request
121                 sample_with_count = worker.sample_with_count.remote()
122                 # Add the task into the sample manager
123                 sample_tasks.add(worker, sample_with_count)
124
125         # Record the time for replay and processing
126         with self.timers["replay_processing"]:
127             for actor, replay in replay_tasks.completed():
128                 # Start another replay task for each completed one
129                 replay_task = actor.replay.remote()
130                 replay_tasks.add(actor, replay_task)
131
```

```python
                # Check the input queue of the learner
                if learner.inqueue.full():
                    num_samples_dropped += 1
                else:
                    # Record the get sample time
                    with self.timers["get_samples"]:
                        samples = ray_get_and_free(replay)

                    # Defensive copy against plasma crashes
                    learner.inqueue.put((actor, samples.copy()))

        # Record the time for priorities update
        with timers["update_priorities"]:
            # Get output from the leaner to update replay priorities on
            # the remote rollout actors and training info
            while not learner.outqueue.empty():
                # Fetch output from the asynchronous learner
                output = learner.outqueue.get()
                actor, priority_dict, count = output

                # Update the priorities on the remote actors
                actor.update_priorities.remote(priority_dict)
                train_timesteps += count

        # Calculate the time information
        time_delta = time.time() - start

        # Collect metrics for training
        timers["sample"].push(time_delta)
        timers["sample"].push_units_processed(sample_timesteps)
        if train_timesteps > 0:
            learning_started = True
        if learning_started:
            timers["train"].push(time_delta)
            timers["train"].push_units_processed(train_timesteps)

        # Update training info
        num_steps_sampled += sample_timesteps
        num_steps_trained += train_timesteps
```