# OpenReview forum: "RLlib Flow: Distributed Reinforcement Learning is a Dataflow Problem"
_NeurIPS.cc/2021/Conference — NeurIPS 2021 Poster_

### Official Review · Reviewer_fzhZ · 2021-06-30

**Rating:** 6
**Confidence:** 4

**Summary:**

It is a good article that makes a significant contribution to the RL domain. It not only identifies the line of thought that hampers the development of RL-based solutions but also proposes the solution. The article makes a comparison of other approaches to their framework to a significant level of detail. Nevertheless, there are some flaws.


**Limitations And Societal Impact:**

Because authors did not address the topic in a theoretical manner, so they were unable to draw a connection between their proposed framework and the established models of parallelism in computer science. This hampered their ability to fully analyze the capacity and limitations of their framework.

**Main Review:**

- A commendable work on reinforcement learning that addresses some important issues
    - The work investigates the limitations of the present RL libraries such as ACME, RLib, and Coach.
    - The work proposes a completely different view on the RL problem and advocates looking at it in a data flow context.
    - It introduces the concept of data flow operators and shows by example how this framework can be used to implement RL-based solutions or experiment with the existing ones

- Despite the good work, there are some flaws in the way article is presented.
    - The framework should have a sound theoretical discussion instead of a discussion about which library to chose for the implementation and which to reject.
    - There are always two main choices to make when implementing a distributed framework
        - Either to use message passing mechanism
        - Or use a shared memory model
    - From a theoretical point of this discussion is important, because it enables us to think above the individual libraries and frameworks.
    - It is well known that the shared memory systems perform better in certain situations as compared to message-passing systems. In other cases, the situation is the opposite. For the sake of completeness, it is essential to have that discussion, whether to implement a data flow-based RL framework using shared memory or message passing.
    - A key benefit of message passing systems is that they are scalable and robust, and can work in many different scenarios and machine capabilities. A shared memory-based system may perform better with greater compute capabilities of machines, but may totally fail in a situation where there are numerous small devices.
- In the light of the above, I think the paper presents a good idea but lacks the theoretical rigor that was needed to draw some long-lasting conclusions.

**Time Spent Reviewing:**

4

---

> ### Author Response · Authors · 2021-08-10
> **Response to Reviewer fzhZ**
>
> Thank you for the wonderful comments. We address your concerns below.
>
> (1) Theoretical discussion.
>
> We would like to emphasize that our AnonFlow is a high-level programming abstraction that can be built on both message passing and shared memory models. The current AnonFlow implementation is based on the distributed shared memory model of Ray, which provides very good performance for RL [1], and, for our purposes, allows for backward compatibility and hence fair direct comparison with RLlib.
>
> One of the advantages of a higher-level programming model like AnonFlow is that it could also be implemented in a shared-memory model. For example, an optimization portion of the dataflow graph could be implemented using mutable shared-memory under the hood, in contrast to Ray's partially message passing and immutable object store. We will add this point to the discussion. We agree this is an interesting topic to explore, but believe it is out of the scope of this paper and will leave it as future work.
>
> [1] Philipp Moritz. "Ray: A Distributed Execution Engine for the Machine Learning Ecosystem".

---

> > ### Author Response · Authors · 2021-09-02
> > **Thank you again for your review Reviewer fzhZ**
> >
> > We would like to thank you again for your review. As the deadline getting close, it is appreciated if you could tell us whether our response addresses your concerns mentioned above and whether it is sufficient to increase the score. Thanks!

---

### Official Review · Reviewer_c8zk · 2021-07-16

**Rating:** 6
**Confidence:** 4

**Summary:**

This paper propose new library for simplifying the implementation of distributed RL algorithms. It identifies the difference between RL workloads and stream processing workloads, and creates a number of abstractions for handling asynchronous parallel operations for RL workloads.



**Limitations And Societal Impact:**

yes

**Main Review:**

The idea is interesting for simplifying the implementation of distributed RL algorithms. It organizes the common distributed actor processing patterns into higher level abstrations.

Although this abstraction simplifies the implementations of existing algorithm, I wonder whether it will have lower flexibiilty for experimentation and new algorithms. For example: dynamically changing dataflow based on the rollout performance.

These abstractions can improve readability of the high-level dataflow, but it seems also hide some of the details of the underlying operations. For example, in Figure 10 b), “ppo_plan” basically hides all the details of how the operations of the PPO algorithm, and how the weights are updated. How do users gain more understanding of the actual operations (like Figure 10a)?

These high level abtractions ease the implementation of distributed RL. However, it increase the overhead when running the algorithm (for example in a single machine). Is it possible to quantify the overhead of these abstractions?

For figure 12, how is the theoretical best performance calculated using Amdahl’s law?

AnonFlow’s performance is very similar to RLlib. Since AnonFlow provides a high level abstraction, does it also allow more lower level optimizations for improving the performance of RL tasks?


**Time Spent Reviewing:**

2

---

> ### Author Response · Authors · 2021-08-10
> **Response to Reviewer c8zk**
>
> Thank you for the insightful comments. The followings are the responses to your concerns.
>
> (1) Flexibility for experimentation and new algorithms
>
> We believe our coverage of multi-agent, model-based, and meta-learning algorithms provide sufficient evidence of AnonFlow's flexibility, as no other major RL libraries support both these and other types of algorithms together in the distributed setting.
>
> (2) It hides some of the details of the underlying operations, e.g. "ppo_plan".
>
> Regarding missing details, we simplified PPO and DQN in the paper to better illustrate how AnonFlow can better combine two flows in the multi-agent training example. The complete plan for PPO and DQN are illustrated below and will be added to the revision:
> ```python
> # ppo subflow
> ppo_op = rollouts.for_each(SelectExperiences(policy="ppo_policy"))
>     .combine(ConcatBatches(min_batch_size))
>     .for_each(TrainOneStep(workers, policy="ppo_policy"))
> ```
>
> ```python
> # dqn subflow
> replay_buffer = create_replay_actors()
> store_op = rollouts.for_each(SelectExperiences(policy="dqn_policy"))
>     .for_each(StoreToBuffer(replay_buffer))
> replay_op = Replay(replay_buffer)
>     .for_each(TrainOneStep(workers, policies="dqn_policy"))
>     .for_each(UpdateTargetNetwork(workers, policy="dqn_policy"))
> dqn_op = Union(store_op, replay_op)
> ```
>
> (3) The overhead of the abstractions on a single machine.
>
> We point the reviewer to Figure 11 (a) and (b), where the 16 and 32 workers settings are running on a single machine. We observe that the overhead introduced by our AnonFlow is negligible.
>
> (4) The calculation of the theoretical best performance by Amdahl's Law.
>
> $f(s_\text{TwoTrainer}) = \frac{1}{\frac{p_\text{PPO}}{s_\text{PPO}}+ \frac{p_\text{DQN}}{s_\text{DQN}}+\frac{p_\text{remaining}}{s_\text{remaining}}} \leq \frac{1}{\frac{p_\text{PPO}}{s_\text{PPO}}+ \frac{p_\text{DQN}}{s_\text{DQN}}+\frac{p_\text{remaining}}{r_\text{workers}}}$
> where $f$ is the theoretical speed up; $p_i$ and $s_i$ are the portion and the speed up of the subtask $i$, respectively, and $r_\text{workers}$ is the ratio of the number of workers after and before scaling up.
>
> (5) Lower-level optimizations.
>
> RL experts can easily customize the execution logic and the data flow separately, thanks to the high modularity of our AnonFlow. Besides, our AnonFlow is an extra abstraction above RLlib, and we preserve the underlying design of basic Ray actors. Therefore, the RL/system experts can also perform low-level optimization using the well-examined Ray APIs.

---

> > ### Author Response · Authors · 2021-09-02
> > **Thank you again for your review Reviewer c8zk**
> >
> > We would like to thank you again for your review. As the deadline getting close, it is appreciated if you could tell us whether our response addresses your concerns mentioned above and whether it is sufficient to increase the score. Thanks!

---

### Official Review · Reviewer_1Njk · 2021-07-18

**Rating:** 6
**Confidence:** 3

**Summary:**

This manuscript proposes AnonFlow, a set of dataflow abstractions that allow for easy specification and implementation of distributed RL algorithms. AnonFlow consists of several composable dataflow operators/iterators implemented on top of RLlib. The manuscript discusses the implementation of several popular RL algorithms and quantifies the effectiveness of AnonFlow in terms of LoC and overall system performance.

**Limitations And Societal Impact:**

The authors addressed the limitations of their implementation.

**Main Review:**

In this paper, the authors combine ideas from dataflow programming with actors and argue that this provides a more flexible programming environment for RL researchers. The methods described in the work, dataflow and actors, are not novel but the combination of these components targeting the development of RL algorithms is interesting. The distinction between the proposed and previous systems, as outlined in Table 1, makes it clear AnonFlow is meant to provide a more generic data distribution scheme alongside a higher-level interface.

Support for the proposed abstractions is provided in the form of examples, in Figures 9 and 10, that use lines of code and readability as the metrics of interest. A reduction w.r.t LOC is expected given the authors positioning of AnonFlow as a high-level interface, relative to RLlib. No performance degradation resulted from the proposed transformations as demonstrated in Figure 11.

The paper is well-written and clearly organized to support the author's RL dataflow/RL argument. Comparisons with competing dataflow and RL libraries are provided to draw attention to the distinctive advantages of AnonFlow.

Overall, although there is a novelty in the expression of the RL algorithms, I think it would be difficult to gauge the significance of the work. It's unclear if the RL community would adopt and/or see value in the insights of the dataflow/actor implementation. I lean toward acceptance because IMHO the current offering of RL frameworks appears to be quite rigid and "new" algorithms typically require a substantial investment in time and effort. From that perspective, this work provides at least one sample of an alternative formulation worth considering to alleviate some concerns.

**Time Spent Reviewing:**

2

---

> ### Author Response · Authors · 2021-08-10
> **Response to Reviewer 1Njk**
>
> Thank you for the encouraging comments. We address the concerns below:
>
> (1) Hard to gauge the significance of the work.
>
> We are not aware of any prior work leveraging streaming dataflow for RL, suggesting that our approach is novel. Beyond identifying that the dataflow model could be a good fit for RL, our paper also (1) provides a concrete proposal as to how this vision can be realized in practice, and (2) demonstrates in a production RL library that it is practically useful.

---

> > ### Comment · Reviewer_1Njk · 2021-08-31
> > **No update**
> >
> > Based on the authors response to my review and the other reviewers I will maintain my current assessment of the manuscript.
> >
> > My comment concerning the "significance" referred more to adoption of the dataflow-centric design pattern by RL developers (it can be ignored since it is a personal opinion and bears no merit on the quality of the provided work). I thank the authors for their response.

---

### Official Review · Reviewer_KAKR · 2021-07-20

**Rating:** 5
**Confidence:** 4

**Summary:**

This paper treats the distributed reinforcement learning as a dataflow problem, and proposes an actor-dataflow programming model named AnonFlow. The author has ported algorithm in RLib to AnonFlow, and reports the throughput comparison among AnonFlow, RLib, Spark and the theoretical optimal performance.

**Limitations And Societal Impact:**

See above.

**Main Review:**

Pros:
+ It is a novel point of view to treat the distributed reinforcement learning as a dataflow problem.

+ This paper proposes AnonFlow, an actor-dataflow programming model for reinforcement learning. Algorithms in RLib has been ported to AnonFlow and the lines of code for distributed execution are reduced.

Cons:
- This paper claims that it is common for RL researchers to eschew existing infrastructures. However, no empirical evidence is provided in the experiment to show that RL researchers prefer to use the proposed AnonFlow, rather than existing infrastructures or writing their own approaches.

- It is insufficient that only evaluating the performance of AnonFlow on several RL algorithms. This paper claims that RL researchers may wish to customize a particular algorithm into various RL tasks (e.g., single-agent, multi-agent, meta-learning) and configuration options (e.g., degree of parallelism, batch size). The author fails to evaluate the performance of AnonFlow on various RL tasks and configuration options.

- The improvement of experimental results is marginal. It can be observed from Figure 11(a) that the sample throughput of AnonFlow on CartPole is less than that of RLib when the number of workers is small. In addition, Figure 11(b) shows that the training throughput of AnonFlow on Atari is almost the same as that of RLib.


**Time Spent Reviewing:**

1

---

> ### Author Response · Authors · 2021-08-10
> **Response to Reviewer KAKR**
>
> Thank you for the great comments. We address the concerns in the following.
>
> (1) Evidence for RL researchers preferring proposed AnonFlow.
>
> It is not easy to quantitatively measure user preference in an open-source project. However, our AnonFlow has already been used in the latest open-sourced distributed RL algorithms, including MAML and multi-agent training, indicating the preference of the RL researchers. As supported in the paper, the AnonFlow design addresses the key user concerns we have identified:
>
> 1. **Higher Level**: people prefer to have a big picture of the algorithm before going deep into the details. In Section 5, we provide numerous implementation examples demonstrating this.
>
> 2. **Lower Code Complexity**: users prefer to write fewer lines of code while not losing readability and flexibility. In Section 6.1, we show AnonFlow reduces code by 1.9-9.6x against the RLlib baseline.
>
> 3. **Good Performance**: In Section 6.2, we compare our AnonFlow with the original RLlib and traditional Streaming system, concluding that the abstraction does not introduce additional overheads to the low-level system (RLlib), and significantly outperforms a general-purpose system (Apache Spark).
>
> (2) Evaluation of Anonflow on various RL tasks.
>
> Thank you for raising the concerns for the performance, but we already evaluated the typical algorithms in our paper, including IMPALA (Figure 11b), PPO (Figure 13) for single-agent training, and Two Trainer Workflow (Figure 12) for multi-agent training. In these experimental results, we also have different configuration options, i.e. the degree of parallelism (by scaling with different #workers).
>
> Since our AnonFlow does not change the underlying algorithm dataflow, just how it is expressed in code, it will not affect the efficiency of the original RLlib with many configuration options, such as batch sizes.
>
> The meta-learning algorithm is the newly supported algorithm thanks to the flexibility of AnonFlow, where we address a common RL task to quickly adapt to new environments. We show tuned experimental results of MAML with our AnonFlow on Ant, Pendulum, and HalfCheetah, where Ant task involves walking to different goals; Pendulum task involves learning how to balance pendulums of different masses; HalfCheetah task involves running either forward or backward. The detailed environment setup, configurations, and logs are in https://anonymous.4open.science/r/MAML-results-for-AnonFLow. We will add these results in our revision.
>
>
> |                            |  Ant  | Pendulum | HalfCheetah |
> | -------------------------- | :---: | :------: | :---------: |
> | #workers                   |  32   |    32    |     20      |
> | Throughput (samples / sec) |  15K  |   24K    |     30K     |
> | Convergence Time           | 3.5h  |   0.3h   |    1.3h     |
>
>
> (3) The improvement is marginal.
>
> We want to emphasize that the goal of AnonFlow is **not** to improve the performance of RL algorithms, but to make it possible to, in the same framework, make complex RL training accessible to RL researchers (Section 2). The comparison indicates that AnonFlow does not introduce performance overheads to a low-level implementation (baseline).

---

> > ### Author Response · Authors · 2021-09-01
> > **Thank you again for your review Reviewer KAKR**
> >
> > We would like to thank you again for your review. As the deadline getting close, it is appreciated if you could tell us whether our response resolve your concerns mentioned above, and whether it changes the position of our paper in your mind. Thanks!

---

### Decision · Program_Chairs · 2021-09-27

**Decision:**

Accept (Poster)

**Comment:**

The reviewers and the AC appreciate this papers contribution of a nice advance in coding machinery to improve processing for parallel streams of RL data. The authors are encouraged to carefully address the multiple suggestions for further strengthening the paper in their camera ready version.